# Performance of different automatic photographic identification software for larvae and adults of the European fire salamander

**Laura Schulte**[1]*, **Charlotte Faul**[2], **Pia Oswald**[1], **Kathleen Preißler**[3], **Sebastian Steinfartz**[3], **Michael Veith**[2‡], **Barbara A. Caspers**[1,4‡]

**1** Department of Behavioural Ecology, Bielefeld University, Konsequenz, Bielefeld, Germany, **2** Biogeography, Trier University, Universitätsring, Trier, Germany, **3** Molecular Evolution and Systematics of Animals, Leipzig University, Talstraße, Leipzig, **4** JICE, Joint Institute for Individualisation in a Changing Environment, University of Münster and Bielefeld University, Bielefeld, Germany

‡ MV and BAC shared the last authorship on this work.
* Schulte.Lra@gmail.com

**Data Availability Statement:** The larval dataset is available here https://doi.org/10.6084/m9.figshare.24999353.v1, and the adult dataset is available

## Abstract

For many species, population sizes are unknown despite their importance for conservation. For population size estimation, capture-mark-recapture (CMR) studies are often used, which include the necessity to identify each individual, mostly through individual markings or genetic characters. Invasive marking techniques, however, can negatively affect the individual fitness. Alternatives are low-impact techniques such as the use of photos for individual identification, for species with stable distinctive phenotypic traits. For the individual identification of photos, a variety of different software, with different requirements, is available. The European fire salamander (*Salamandra salamandra*) is a species in which individuals, both at the larval stage and as adults, have individual specific patterns that allow for individual identification. In this study, we compared the performance of five different software for the use of photographic identification for the European fire salamander: Amphibian & Reptile Wildbook (ARW), AmphIdent, I3S pattern+, ManderMatcher and Wild-ID. While adults can be identified by all five software, European fire salamander larvae can currently only be identified by two of the five (ARW and Wild-ID). We used one dataset of European fire salamander larval pictures taken in the laboratory and tested this dataset in two of the five software (ARW and Wild-ID). We used another dataset of European fire salamander adult pictures taken in the field and tested this using all five software. We compared the requirements of all software on the pictures used and calculated the False Rejection Rate (FRR) and the Recognition Rate (RR). For the larval dataset (421 pictures) we found that the ARW and Wild-ID performed equally well for individual identification (99.6% and 100% Recognition Rate, respectively). For the adult dataset (377 pictures), we found the best False Rejection Rate in ManderMatcher and the highest Recognition Rate in the ARW. Additionally, the ARW is the only program that requires no image pre-processing. In times of amphibian declines, non-invasive photo identification software allowing capture-mark-recapture studies help to gain

here https://doi.org/10.6084/m9.figshare.24998690.v1.

**Funding:** This research was conducted as part of the CRC TRR 212 (NC³) – Project number 316099922 and 396777092 funded by the German Research Foundation (DFG). The funders had no role in study design, data collection and analysis, decision to publish, or preparation of the manuscript.

**Competing interests:** The department of Behavioural Ecology from Bielefeld University placed the order for the Amphibian & Reptile Wildbook software together with NC3 (Collaborative research centre "A Novel Synthesis of Individualisation across Behaviour, Ecology and Evolution: Niche Choice, Niche Conformance, Niche Construction (NC3)" founded by the DZG) and Heike Pröhl and Mirjam Nadjafzadeh, who received funding from the Deutsche Gesellschaft für Herpetologie und Terrarienkunde (DGHT). They are not receiving any kind of payment or other benefits in relation with the Amphibian & Reptile Wildbook." This does not alter our adherence to PLOS ONE policies on sharing data and materials.

knowledge on population sizes, distribution, movement and demography of a population and can thus help to support species conservation.

## Introduction

Worldwide, 41% of the amphibian species suffer from population declines, and the conservation status of an additional 11% is not even known due to deficient data [1–3]. Thus, knowledge of the ecology of these species, including population size, is crucial for implementing conservation measures [4]. Capture-mark-recapture (CMR) studies are among the most used methods to estimate abundance of animals, to track life-histories, give information on growth rate, and movements of individuals and to monitor the dynamic of communities and are a useful tool to estimate population sizes [5–9].

Reliable population size estimates based on CMR studies require the recognition of an individual and thus appropriate marking techniques, which depend on the size, life stage and the studied species itself. For instance, RFID tags can be used for insects, radio transmitters or rings for birds, passive integrated transponder (PIT) tags in crustaceans and tattoos for mammals [6, 10, 11]. Furthermore, such markings should be permanent or at least last for the duration of the experiment [10, 12]. However, often high costs are included for invasive tags [11], which is an issue in both ecological research and conservation projects.

Due to their permeable and sensitive skin, amphibians are difficult to tag [13, but see e.g. 8 for a study on PIT tags inserted in the skin). Particularly marking amphibian larvae is challenging and there are only few techniques possible, such as fin clipping or staining [reviewed in 10]. Other invasive marking techniques for amphibians, such as toe clipping, do not only concern ethical issues [14] and increase handling time, they also increase stress for the organism [15], may enhance infections, can reduce individual fitness and may ultimately affect survival rates [10, 16], thus confounding population size estimates. Alternatively, PIT tags, inserted under the skin, have been proven to have no negative effect on adult European fire salamanders (*Salamandra salamandra*), making them a useful tool for marking species with a less distinct or no colour pattern, such as the alpine salamander (*S. atra*) [8]. Also, Visual Implant Elastomers (VIE) marks were recovered successfully over a period of five years of individually marked Italian cave salamander (*Hydromantes italicus*) [17]. In contrast, the use of Visible Implant (VI) Alpha tags in larvae of the European fire salamander negatively influenced their fitness [18].

Photographic identification is a widely acknowledged non-invasive marking method. It provides an ideal way to recognize individuals without the potential negative effects of invasive markings [19, 20] and was already successfully used in a variety of amphibian species [21]. Moreover, natural marks and patterns of animals usually cannot get lost, moulted or shed [21, 22], although some animals have patterns that change over time with age such as in Grass Snakes (*Natrix helvetica*) [23]. To enhance photo-identification, the use of photo recognition software has been proven useful. It is faster than the inspection by eye, less error-prone and also less prone to observer bias [5, 19, 24]. It allows handling large datasets [25, 26] and can be less expensive than other marking techniques [27]. However, the specific pattern of a species needs to be tested first, if it is reliable over time to use for individual identification [28].

European fire salamanders show an individual-specific pattern, both during the larval stage as well as after metamorphosis, which allows individual identification [29–32]. The European fire salamander is widely distributed across Europe and inhabits preferably deciduous forests

with access to water bodies for larval deposition [33]. In Europe, the species is listed as Least Concern [34]. However, in Germany the European fire salamander is now listed on the pre-warning list [35]. The spread of the chytrid fungus *Batrachochytrium salamandrivorans* (*Bsal*) in Europe and particularly in Germany might further negatively affect European fire salamander populations [36]. Therefore, populations of the species are increasingly monitored [e.g., 9, 18, 37].

The performance of photographic identification has already been evaluated in several studies. These studies compared the between-observer variability when matching by eye vs. recognition software [19], tested the performance of different matching algorithms [e.g. 31], or a completely different data source and matching algorithm [e.g. genetic data; 38]. Their findings suggest that the use of photographic matching software is better than matching by eye and that the algorithm used by a software strongly influences the matching results. The Amphibian and Reptile Wildbook (ARW) is a recently launched new pattern recognition software which is at the moment available for adults of the Yellow-bellied toad (*Bombina variegata*) and adult as well as larval pictures of the European fire salamander and the Near-eastern fire salamander (*S. infraimmaculata*). In contrast to other recognition software, it aims at a broad application in citizen science projects and constitutes an online option for the analysis of images of European fire salamanders that have not been pre-processed. The performance of the programme has not yet been tested in comparison to existing photo-identification software. We therefore use one adult and one larval European fire salamander dataset to compare software for individual pattern recognition. The adult dataset was tested on five software, whereas the larval dataset was tested on two software only due to software limitations and picture requirements. We used the following five software programmes: I3S and Wild-ID, as they were already commonly used for CMR studies [e.g. 39, 40], and ARW, AmphIdent and ManderMatcher, which were especially developed for amphibian studies or studies on European fire salamanders [32, 41]. The aim of this study was to compare the different software by comparing the Recognition Rate (RR), False Rejection Rate (FRR) and the pre-processing requirements.

## Methods

### Compared software

**Amphibian & reptile wildbook.** ARW is a free and web-based software that was released in 2021 (https://amphibian-reptile.wildbook.org). It can be accessed online anywhere at any time. If the users upload their pictures publicly, it allows them to compare own captures with captures of other users that might have encountered the same individual [42]. In this way, citizen scientists can contribute to research and conservation with their encounters of animals by enriching an existing database. Currently, ARW is validated for adults of Yellow-bellied toads, as well as larvae and adults of the European fire salamander and the Near eastern fire salamander. ARW does not require processing of the uploaded images (Table 1), as the Hotspotter algorithm will automatically detect the position of the animal in the picture. It has recently been successfully used for a population size analysis of European fire salamander larvae in Germany [9].

**AmphIdent.** AmphIdent is a commercial standard program for the identification of individuals of amphibian species. Different modules have been developed, such as for the great crested newt (*Triturus cristatus*), the European fire-bellied toad (*B. bombina*), the Yellow-bellied toad, the European fire salamander and the Marbled salamander (*Ambystoma opacum*). The software can be purchased at www.amphident.de. It requires the pre-processing of each picture as the body part to be compared needs to be chosen from the picture (e.g. the dorsal pattern of adult European fire salamanders). AmphIdent uses a pixel-based algorithm which

**Table 1. Image processing requirements and species availability for each software.**

|  | Image processing required? | Available for which species? |
|---|---|---|
| Amphibian & Reptile Wildbook (ARW) | No | Yellow-bellied toad (*Bombina variegata*), European fire salamander (*Salamandra salamandra*), Near-eastern fire salamander (*S. infraimmaculata*) |
| AmphIdent | Yes | Great crested newt (*Triturus cristatus*), European fire-bellied toad (*B. bombina*), the Yellow-bellied toad, European fire salamander, the Marbled salamander (*Ambystoma opacum*) |
| I3S pattern+ | Yes | Every species with individual pattern |
| ManderMatcher | Yes | Fire salamander species |
| Wild-ID | Recommended | Every species with individual pattern |

was proven to be very robust with a large dataset in comparison to other recognition software [31, 38].

**Interactive individual identification system (I3S).** I3S is a freely available software (https://reijns.com/i3s/) and was originally developed for sharks [43]. It now offers a suite of software solutions depending on the individual marking of the study species, e.g. spots for I3S classic. In each software, image processing is needed as the user needs to choose reference points in each image [43]. I3S uses the Scale Invariant Feature Transform (SIFT) algorithm which has been found to be very stable and repeatable regarding its performance [21]. It was successfully used for different species, such as two European lizard species [4], and I3S pattern works well for amphibians with large spots such as the Italian crested newt (*Triturus carnifex*) [21]. For this study, I3S pattern+ was chosen due to suitability to recognise the striped pattern.

**ManderMatcher.** ManderMatcher is a free and pattern based tool (https://jeroenspeybroeck.shinyapps.io/mandermatcher/) that was successfully applied for *S. salamandra* but can also be used for other *Salamandra* species such as *S. algira*, *S. corsica* and *S. infraimmaculata* [32]. Image processing is required as every image needs to be coded after the salamanders body part, e.g. 11 characteristics need to be coded for the head [32].

**Wild-ID.** Wild-ID is a free available software (https://faculty-directory.dartmouth.edu/douglas-thomas-bolger) that uses the SIFT operator. This operator detects characteristic image features that can then be compared over a large dataset [5, 44]. Wild-ID can be used for all species that show a distinctive colour pattern [19]. For better matching results, the software requires each picture to be cropped to the area of interest [5]. Wild-ID has been successfully used to study amphibians before, e.g., the Jollyville Plateau salamander (*Eurycea tonkawae*) and the Lake Oku clawed frog (*Xenopus longipes*) [12, 19]. In particular, even amphibian larvae can be identified, for example by comparing the pattern on the tail fin pattern of larvae of the European fire salamander [30].

### Used datasets

**Larval dataset & mirrored larvae test.** In the study of Faul et al. (2022), 40 European fire salamander larvae were captured in the Hunsrück Mountains (Germany) and kept under standardized conditions for up to ten weeks until the onset of metamorphosis. Starting with the second week, each larva was photographed under standardized conditions on a weekly basis [for further details see 30]. Hence, from week 2 on, every picture mimics a recapture event. We used the pictures from Faul et al. (2022) from the left body side (291 pictures, including 251 recaptures, Table 2) to compare software performance. Like Faul et al. (2022), we included 130 additional pictures of wild caught larvae from the same creek to the dataset, as the performance of photo-matching software is known to decrease with an increasing dataset [31]. Due to the specific spots on the tail fin, only Wild-ID and ARW can process the pictures and match

**Table 2. Tested datasets and software used per dataset.**

| Dataset | Number of pictures per dataset | Software used |
|---|---|---|
| Larval dataset | 421 | ARW, Wild-ID |
| Mirrored larvae test | 30 | ARW, Wild-ID |
| Adult dataset | 377 | ARW, AmphIdent, I3S pattern+, ManderMatcher, Wild-ID |

individuals among the compared software. For the use in Wild-ID, all pictures have been pre-processed by Faul et al. (2022) in a way that the part of the tail from the base of the hind leg to the end of the tail was cut out in a ratio 10:3.5. For the use in the ARW software, the pictures were not pre-processed.

For the mirrored larvae test, we have used a subset of the larval dataset. Faul et al. (2022) identified individual larvae by using only photos from the same body side. However, this can be time-consuming, especially in in the field, and cause additional stress for the larvae as handling time increases. Since larval tails are translucent and parts of the pattern of one side are always visible from the respective other side, we tested whether larvae could be reliably identified even when the photographed side of the larvae was not the same. We selected a random subset of 30 pictures from Faul et al. (2022) from the other body side (here the right one). We digitally mirrored these images (Fig 1) and tested if the ARW and Wild-ID were able to find correct matches among the complete dataset of the images of the left body sides.

## Adult dataset

Between 2018 and 2020, 377 pictures of adult European fire salamanders were taken in the Kruppwald forest in the city of Essen (Germany). These are images of a population infected with *Bsal*. Images were taken from the dorsal side under non-standardized conditions in the field with varying backgrounds, such as forest floor, graph paper and sometimes even in the hand leading to an inhomogeneous dataset with different qualities of the pictures. The total dataset included 66 recaptures (17.5% recapture rate), identified via visual inspection in combination with the results of all software tested here. For visual inspection, two people manually compared all the pictures and assigned the recaptures by eye.

## Recognition rate (RR) & false rejection rate (FRR)

To quantify software performance, we calculated the Recognition Rate (%) and the False Rejection Rate (%) for each dataset and for each software separately. We defined the Recognition Rate as the proportion of recaptures from the full dataset that was successfully identified as a match by the software. For the larval dataset in Wild-ID, we report the existing FRR from Faul

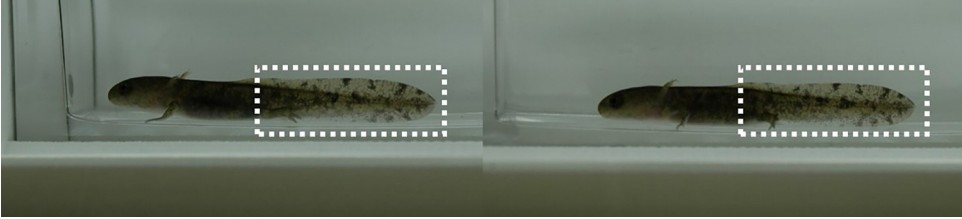

**Fig 1. Two pictures of the same individual of a European fire salamander larva taken in week two of the experiment.** Left: Picture taken from the left body side of the individual. Right: Picture taken from the right body side of the individual and then mirrored on the computer. White dashed boxes show the tail in which the upper and lower part are translucent whereas the centre is not translucent (pictures: Charlotte Faul).

et al. (2022). The FRR describes the probability of failing to recognise two pictures of the same individual within a given dataset [5] and was calculated following Bolger et al. (2012) and Faul et al. (2022). We used it in three versions [5, 30]: FRR1 refers to cases where a software was unable to place a true match in the first position; FRR10 refers to cases where no true matches were placed among the ten highest ranking images; FRR21 refers to cases where a software was unable to assign a correct image of the same individual to a test image within the 20 potential matches. Thus, a high FRR value indicates poor recognition, while a low FRR indicates successful recognition of two pictures of the same individual. All software software require the observer to confirm or reject the proposed matches.

## Results & discussion

### Larval dataset & mirrored larvae test

ARW and Wild-ID performed almost equally well when matching larval pictures, regarding both the RR and the FRR (Table 3). ARW was not able to match two pictures of the same amelanistic larva, while Wild-ID was able to match the pictures.

The FRR showed some dissimilarity between Wild-ID and ARW. We found a higher FRR1 for Wild-ID compared to ARW (3.52% and 0%, respectively), while again the FRR10 and the FRR21 were almost identical for Wild-ID and ARW (Table 3). All FRR values from both software can be considered very low and thus indicate a reliable image detection. Another study compared the tail fin venation of amphibian larvae by untrained citizen scientist by eye. Here, they demonstrated that the consensus of all participants on matches of *Litoria aurea* tadpoles varied between 67% and 83% [45] and thus showing a lower Recognition Rate in comparison to the software indicating that the use of a software outcompetes by eye matching of (untrained) citizen scientists. Hence, both software reflect a reliable and beneficial tool in European fire salamander larvae research and conservation.

Furthermore, we tested the performance of ARW and Wild-ID by applying mirrored larval images to both software applications. In both cases, we found a very low FRR and a very high RR, the latter being 96.67% and 100%, respectively (Table 3). This shows that both applications reliably match an image of one side of the body to an image of the other side of the body, which is important to have in mind when handling of larvae during fieldwork. Pictures of either body side can be taken in the field; this reduces stress to the individuals and time of field work. However, the time for subsequent processing of the images on the computer is increased.

### Adult dataset

In our adult dataset, we found great differences among the software we tested. AmphIdent and I3S pattern+ performed comparably well, with the RR being higher for AmphIdent and the

**Table 3. Recognition rate (RR, %) and false rejection rates FRR1, FRR10 and FRR21 (all in %) for the larval European fire salamander datasets.**

| Dataset | RR | FRR1 | FRR10 | FRR21 |
|---|---|---|---|---|
| *Larvae* | | | | |
| ARW | 99.60 | 0.24 | 0.24 | 0.24 |
| Wild-ID | 100.00 | 3.52 | 0.00 | 0.00 |
| *Mirrored larvae test* | | | | |
| ARW | 96.67 | 0.01 | 0.00 | 0.00 |
| Wild-ID | 100.00 | 0.24 | 0.00 | 0.00 |

**Table 4. Recognition rate (RR, %) and false rejection rates FRR1, FRR10 and FRR21 (all in %) for the adult datasets; the combination of visual inspection and software results was used to define 100% of recaptures.**

| Dataset | RR | FRR1 | FRR10 | FRR21 |
|---|---|---|---|---|
| ARW | 77.27 | 10.88 | 6.90 | 3.98 |
| AmphIdent | 60.61 | 12.47 | 8.22 | 7.43 |
| I3S pattern+ | 56.06 | 10.61 | 7.69 | 7.69 |
| ManderMatcher | 72.73 | 9.81 | 5.31 | 5.04 |
| Wild-ID | 37.88 | 15.12 | 12.47 | 10.88 |
| Visual inspection | 93.94 | - | - | - |
| Visual & Software | 100.00 | - | - | - |

FRR1 and the FRR10 being lower for I3S pattern+ (Table 4). Both software have also been used very successfully in other studies on amphibians and reptiles [e.g., 4, 21, 31]. However, in a study on Great crested newts, Drechsler et al. (2015) found a low FRR of only 2% for AmphIdent. Nevertheless, Drechsler et al. (2015) also showed that pictures with poor quality (e.g., blurriness) could not be processed with AmphIdent and needed to be excluded from their study. While AmphIdent is available for only a few species, I3S is offering a suite of software solutions for species with different types of patterns. Goedbloed et al. (2017) demonstrated that AmphIdent produces much smaller FRR values for standardized (FRR = 0%) than for non-standardized (FRR = 35.2%) pictures, suggesting to always take standardized pictures with specific angles. Similarly, for I3S, Sacchi et al. (2010) found a high RR of 98%-99% in a lizard dataset; they had captured each lizard and photographed the ventral side of the body, thus also creating a highly standardized dataset. This can however be challenging during fieldwork if the animals are not captured as they often move or can be partially hidden by vegetation, which is hard to control. Thus, additional handling may be needed.

While Wild-ID performed well for the larval dataset, performance for the adult dataset was much lower as indicated by the highest False Rejection Rates found in our study for FRR1, FRR10 and FRR21 (15.12%, 12.47% and 10.88%, respectively) and the lowest RR values with only 37.88%. In line with our results, Matthé et al. (2017) also found an RR value of only 30–49% when Wild-ID was used to analyse a European fire salamander dataset. In our study, we had trimmed all pictures prior to matching to increase the matching probability [5]. One explanation for this low performance could be that the adult images, unlike the larval images, were not taken under standardized conditions, and although the implemented SIFT operator should account for rotations and different views [44], in most cases the animal's positions may have been too variable for successful matching. In previous studies, Wild-ID has been demonstrated to match images very successfully [e.g., 5, 25, 46], including those with non-standardized images from Thornicroft's giraffe (*Giraffa camelopardalis thornicrofti*) for instance [39]. Halloran et al. (2015) found that background complexity was the only factor that negatively affected matching success, rather than animal orientation. However, as their study was conducted with giraffes, they photographed the animals from the side and not from above like adult European fire salamanders. When European fire salamanders are photographed from above, their bodies may be bended or stretched, as it is the case in our dataset. This is likely to have reduced the performance of Wild-ID.

We found the lowest FRR1 and FRR10 values in ManderMatcher (9.81% and 5.31%, respectively) and the lowest FRR21 value in ARW (3.98%). ARW found 77.3% of recaptures, which is the highest Recognition Rate among the software compared. However, other studies reported Recognition Rates as high as 98%-99% and 100% (using e.g., Wild-ID 23; or I3S 4). Bendik et al. (2013) found that the quality of the dataset plays an important role as they demonstrated

that the FRR for a good quality dataset was 20 times lower than for a poor-quality dataset. However, in our adult dataset the quality differed substantially between the pictures as they were taken under non-standardised conditions in the field, but both software were able to match pictures reliably.

All software tested suggest potential matches and then require confirmation of a match by the users eye. However, even the best performing software, the ARW, was only able to identify 77.3% of all recaptures. For our dataset, which contained an *a priori* unknown number of recaptures, we found that the combination of visual and automated inspection led to the highest Recognition Rate. Visual inspection resulted in an RR of 93.94%, and we defined the combination of visual and computer-aided inspection to be 100% of all recaptures. Visual inspection is by far the most time-consuming method, which can hardly be applied to large datasets. Another possibility to improve reliability of the software could be to combine results from two software programmes, like Wild-ID and ARW for instance. However, different software have different requirements for the image processing and using more than one software can also be considered quite time consuming and thus not very effective.

In summary, time efficient and less error-prone automated photographic identification of individuals is an indispensable tool for ecological research and conservation biology while one needs to assume that in non-standardised datasets not 100% of all actual recaptures will be found.

## Conclusion

Taking images under standardized conditions plays a major role in the success of recognition software. In contrast to adult images, pictures of larvae will always be taken under more or less standardized conditions, since each larva has to be removed from its water body in order to take a picture of the side of the body as it was the case in our study. This might explain why both ARW and Wild-ID performed better in matching larval than adult images, as the latter were not taken in a standardized way. We assume that (especially in citizen science projects) most pictures of adult European fire salamanders are not taken under standardized conditions, as the animals often move when approached and therefore many images are likely to be taken opportunistically rather than in a controlled manner.

The newly launched ARW software offers several benefits in the picture matching process. It performed very well for the larval dataset and showed the best recapture rate for the adult dataset, thus users only need to use one software to process images of both life stages. Moreover, it is the only software (among the compared) that does not require any image pre-process prior to detect matches. In this way, ARW can be easily used for small and large datasets. In addition, it is likely to deliver even better results in the future as the hot spotter algorithm gets better ("learns") with more images being matched. Given that it is available online, it also offers the opportunity to involve citizen scientist. In conclusion, the time efficiency that ARW offers and considering the good performance of the software, it is a helpful tool that can be used for the monitoring of larval and adult European fire salamander populations and thus serve species conservation.

## Acknowledgments

We thank Lara Gemeinhardt, Marvin Krebs, Julian Enß, Marine Klamke, Felix Kamprad and Alina Schulz who were involved in taking the pictures of adult fire salamanders in the Kruppwald (permission number 59-5-1-53 (1398), Untere Naturschutzbehörde, Essen) as well as the students that helped checking for recaptures by eye and with software.

## Author Contributions

**Conceptualization:** Laura Schulte, Pia Oswald, Kathleen Preißler, Sebastian Steinfartz, Michael Veith, Barbara A. Caspers.

**Data curation:** Charlotte Faul, Kathleen Preißler.

**Formal analysis:** Laura Schulte, Charlotte Faul.

**Funding acquisition:** Barbara A. Caspers.

**Methodology:** Laura Schulte, Michael Veith, Barbara A. Caspers.

**Software:** Barbara A. Caspers.

**Supervision:** Michael Veith, Barbara A. Caspers.

**Writing – original draft:** Laura Schulte.

**Writing – review & editing:** Charlotte Faul, Pia Oswald, Kathleen Preißler, Sebastian Steinfartz, Michael Veith, Barbara A. Caspers.

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
