## [Decision Letter · Decision Letter 0]

5 Dec 2023

PONE-D-23-31936Performance of different automatic photographic identification software for larvae and adults of the European fire salamanderPLOS ONE

Dear Dr. Schulte,

Thank you for submitting your manuscript to PLOS ONE. After careful consideration, we feel that it has merit but does not fully meet PLOS ONE’s publication criteria as it currently stands. Therefore, we invite you to submit a revised version of the manuscript that addresses the points raised during the review process.

Based on the comments from two reviewers and my own reading, I suggest that the authors can make some minor revisions to improve the manuscript. In particular, Reviewer 2 made some good comments.

We look forward to receiving your revised manuscript.

Kind regards,

Christopher Nice, Ph.D.

Academic Editor

PLOS ONE

“This research was conducted as part of the CRC TRR 212 (NC³) – Project number 316099922 and 396777092 funded by the German Research Foundation (DFG).”

“The department of Behavioural Ecology from Bielefeld University placed the order for the Amphibian & Reptile Wildbook software together with NC3 (Collaborative research centre “A Novel Synthesis of Individualisation across Behaviour, Ecology and Evolution: Niche Choice, Niche Conformance, Niche Construction (NC3)” founded by the DZG) and Heike Pröhl and Mirjam Nadjafzadeh, who received funding from the Deutsche Gesellschaft für Herpetologie und Terrarienkunde (DGHT). They are not receiving any kind of payment or other benefits in relation with the Amphibian & Reptile Wildbook.”

6. Please remove your figures from within your manuscript file, leaving only the individual TIFF/EPS image files, uploaded separately. These will be automatically included in the reviewers’ PDF.

Reviewers' comments:

Reviewer's Responses to Questions

**Comments to the Author**

1. Is the manuscript technically sound, and do the data support the conclusions?

Reviewer #1: Yes

Reviewer #2: Yes

2. Has the statistical analysis been performed appropriately and rigorously? 

Reviewer #1: N/A

Reviewer #2: Yes

3. Have the authors made all data underlying the findings in their manuscript fully available?

Reviewer #1: No

Reviewer #2: No

4. Is the manuscript presented in an intelligible fashion and written in standard English?

Reviewer #1: Yes

Reviewer #2: Yes

5. Review Comments to the Author

Reviewer #1: This is a nice and well-written manuscript where authors compare the accuracy of four softwares for individual recognition of species with well defined pattern, in the specific case for the fire salamander. I find it interesting but with a too narrow scope for this journal. I would recommend to target a more sectorial journal, especially because the whole study is focused on a single species. Furthermore, the used dataset is missing as supplementary material

Reviewer #2: Schulte et al. describe a study in which they compare a series of programs used in automatic photographic identification for photo-mark-recapture techniques. I find this topic to be a valuable contribution to the field, and I think the authors describe their results succinctly and relatively clearly. My only major reservation is that the authors never define “Recognition Rate”, which makes some of the results uninterpretable. I think that this is something that the authors could easily fix, though. Below are some more minor line-by-line comments.

Line 26: There is a mismatch in singular vs. plural here. I would recommend changing it to “An alternative can be found among non-invasive techniques…” or something like that.

Line 32: I think that “computer-based” is redundant and unnecessary here.

Lines 34–35: As I understand it, this is a current limitation of this software. But it may not be permanent. So I would say, “…larvae can currently only be identified…”. Perhaps new modules will become available in the future, so to make sure this study remains accurate through time, I would encourage the authors to rephrase this.

Line 36: Change “larvae” to “larval”. In general, this manuscript is well-written and the English is clear. There are several places where it could use some minor improvement, but I do not view this as a barrier to publication. If the authors are interested, the Society for the Study of Amphibians and Reptiles does offer a Manuscript Review Service for this purpose (see details here: https://ssarherps.org/publications/manuscript-review-service/).

Line 51: There is an updated analysis that was recently published on this topic. “Ongoing declines for the world’s amphibians in the face of emerging threats” by Luedtke et al. (2023). The authors may wish to cite this for the latest information.

Line 61: Unless I am misunderstanding, I think these are more often referred to as “bands” than “rings”.

Lines 119–120: It is not technically wrong to say “…the following five software”, but it would be more common in English to say “…the following five software programs”, even if it is unnecessarily redundant in meaning. Just an idiosyncrasy of the language. I don’t think a change here (and in other places throughout the manuscript where this comment is relevant) is necessary, but the authors may want to consider it.

Lines 182–184: I think the authors should consider citing “Computer-assisted photo identification outperforms visible implant elastomers in an endangered salamander, Eurycea tonkawae” by Bendik et al. (2013) as another example of Wild-ID being applied to salamanders. Perhaps the information in the discussion of this paper would also prove valuable.

Line 186: As the authors describe earlier, image processing is recommended for Wild-ID, but it is not required. This should be clearer in the table.

Line 231: Recognition Rate should also be defined clearly here. The results are hard to interpret otherwise.

Line 233: Does “we used the existing FRR from Faul et al. (2022)” mean that the authors simply adopted the definitions from this paper, but recalculate these values? Or that they simply report the exact same values originally calculated in that study?

Line 256: Should this say “FRR” instead of “FFR”? Likewise on Lines 257, 267, and several other places.

Line 296: As noted earlier, I think it is critical that “Recognition Rate” (RR) is defined. I would have assumed that it means something like “the probability of identifying a true match somewhere in the list of potential matches”. However, this doesn’t appear to be possible. For example, an FRR1 of 15.12 in Wild-ID would suggest that the correct match *was* identified in the first position for ~85% of photographs. However, the RR is only 37.88. This seems to mean that my intuition is wrong about the definition of RR. I think that a better explanation of this should be a priority.

Line 337: I think the authors may consider expanding their discussion in two ways: 1) Would there be value in combining several of these methods? The authors acknowledge that manual visual inspection + software is very effective. But what about, for example, ARW + Wild-ID? Would that improve the ability to do this reliably?; and 2) This seems like a field that is likely to change quickly and dramatically with the advent of modern artificial intelligence tools. Is there reason to believe that all of these existing tools (i.e., the ones compared in this study) might be replaced in the near future? I am not asking the authors to speculate wildly, but a brief discussion of the future directions for these methods might prove useful.

Line 374: This does not appear to satisfy the PLOS Data policy. I recommend making these underlying data available in a public repository.

6. PLOS authors have the option to publish the peer review history of their article (what does this mean?). If published, this will include your full peer review and any attached files.

Reviewer #1: No

Reviewer #2: No

---

## [Author Response · Author response to Decision Letter 0]

16 Jan 2024

Dear Editor and Reviewers, 

Thank you very much for your useful feedback on our manuscript. We appreciate your time and effort to improve this manuscript. 

Please find below in red our responses to your comments with the corresponding lines from the updated manuscript (with track changes). 

Reviewer #1

L26 tautological maybe change with "Alternative low-impact techniques such as..."

We’ve changed it, thanks. L 25 “Alternatives are low-impact techniques such as the use of photos for individual identification, for species with stable distinctive phenotypic traits.”

L27 please rephrase with something more concise, like "...species with stable distinctive phenotypic traits"

We’ve changed it, thanks. Please see previous comment.

L53 I would say are among the most used methods, as recently an even less invasive method demonstrated to be highly reliable to estimate populations size, even without handling individuals: repeated counts. This method is even easier to use than CMR. I would then suggest to highlight more the advantages that CMR can give, such as information on growth rate, home ranges...

Ficetola, G. F., B. Barzaghi, A. Melotto, M. Muraro, E. Lunghi, C. Canedoli, E. Lo Parrino, V. Nanni, I. Silva-Rocha, A. Urso, M. A. Carretero, D. Salvi, S. Scali, G. Scarì, R. Pennati, F. Andreone and R. Manenti (2018). "N-mixture models reliably estimate the abundance of small vertebrates." Scientific Reports 8: 10357.

We’ve rephrased it, thanks. L 51 “Capture-mark-recapture (CMR) studies are among the most used methods to estimate abundance of animals, to track life-histories, give information on growth rate, and movements of individuals and to monitor the dynamic of communities and are a useful tool to estimate population sizes (5–9).”

L79 The Authors should mention also the use of probably the best tagging method, visual implant elastomers, widlely used on frogs and salamanders species, which can last for years and can be also retained after larvae metamorphosis

Grant, E.H.C. (2008). Visual implant elastomer mark retention through metamorphosis in amphibian larvae. Journal of Wildlife Management 72:1247–1252.

Lunghi, E. and G. Bruni (2018). "Long-term reliability of Visual Implant Elastomers in the Italian cave salamander (Hydromantes italicus)." Salamandra 54(4): 283-286.

We’ve added the study, thanks. However, we do not agree with considering it the best tagging method. In Lunghi & Bruni (2018) they could only recover 4 out of 77 tagged salamanders. Indeed, the 4 were not harmed but it remained unknown what happened to the rest so I would argue that it cannot be ruled out that individuals died because of the marking or the wounds that the markings left. 

L 71 “Also, Visual Implant Elastomers (VIE) marks were recovered successfully over a period of five years of individually marked Italian cave salamander (Hydromantes italicus)(17).”

L92 It should be mentioned that its efficiency is limited to well-defined patterns. Indeed, as an example there are European cave salamanders: both ventral and dorsal pattern may be used for individual recognition, but only the former can be done using softwares

Lunghi, E., D. Romeo, M. Mulargia, R. Cogoni, R. Manenti, C. Corti, G. F. Ficetola and M. Veith (2019). "On the stability of the dorsal pattern of European cave salamanders (genus Hydromantes)." Herpetozoa 32: 249-253.

Renet, J., L. Leprêtre, J. Champagnon, and P. Lambret. 2019. Monitoring amphibian species with complex chromatophore patterns: a non-invasive approach with an evaluation of software effectiveness and reliability. Herpetological Journal 29:13–22.

We added the first study, thanks. L 82 “However, the specific pattern of a species needs to be tested first, if it is reliable over time to use for individual identification (1).“

Table1 maybe tested on? otherwise it seems that you can employ dataset only of the mentioned species

So far, only the mentioned species are available for the named software programmes so we guess “availability” is a proper term here. The ARW for example can right now only be used for three species. It can be expanded to other species but that would require programming and training the software first. 

Reviewer #2

Line 26: There is a mismatch in singular vs. plural here. I would recommend changing it to “An alternative can be found among non-invasive techniques…” or something like that.

We’ve already changed the sentence according to the recommendation of the other reviewer. L 25 “Alternatives are low-impact techniques such as the use of photos for individual identification, for species with stable distinctive phenotypic traits.”

Line 32: I think that “computer-based” is redundant and unnecessary here.

We’ve removed it, thanks. 

Lines 34–35: As I understand it, this is a current limitation of this software. But it may not be permanent. So I would say, “…larvae can currently only be identified…”. Perhaps new modules will become available in the future, so to make sure this study remains accurate through time, I would encourage the authors to rephrase this.

We’ve changed it, thanks. L 33 “While adults can be identified by all five software, European fire salamander larvae can currently only be identified by two of the five (ARW and Wild-ID).”

Line 36: Change “larvae” to “larval”. In general, this manuscript is well-written and the English is clear. There are several places where it could use some minor improvement, but I do not view this as a barrier to publication. If the authors are interested, the Society for the Study of Amphibians and Reptiles does offer a Manuscript Review Service for this purpose (see details here: https://ssarherps.org/publications/manuscript-review-service/).

We’ve changed it, thanks.

Line 51: There is an updated analysis that was recently published on this topic. “Ongoing declines for the world’s amphibians in the face of emerging threats” by Luedtke et al. (2023). The authors may wish to cite this for the latest information.

We’ve added this study, thanks. L 49 “Worldwide, 41% of the amphibian species suffer from population declines, and the conservation status of an additional 11% is not even known due to deficient data (1–3).” 

Line 61: Unless I am misunderstanding, I think these are more often referred to as “bands” than “rings”.

In both north and south America, it is indeed referred to as “bands”, whereas here in Europe it is called bird “ringing”. As the fire salamander is distributed here, we would like to stick with the “European” term. 

Lines 119–120: It is not technically wrong to say “…the following five software”, but it would be more common in English to say “…the following five software programs”, even if it is unnecessarily redundant in meaning. Just an idiosyncrasy of the language. I don’t think a change here (and in other places throughout the manuscript where this comment is relevant) is necessary, but the authors may want to consider it.

We’ve added it, thanks. L 108 “We used the following five software programmes”

Lines 182–184: I think the authors should consider citing “Computer-assisted photo identification outperforms visible implant elastomers in an endangered salamander, Eurycea tonkawae” by Bendik et al. (2013) as another example of Wild-ID being applied to salamanders. Perhaps the information in the discussion of this paper would also prove valuable.

We’ve added the study, thanks. L 163 “Wild-ID has been successfully used to study amphibians before, e.g., the Yellow-bellied toad, the Jollyville Plateau salamander (Eurycea tonkawae) and the Lake Oku clawed frog (Xenopus longipes) (12,19,27).”

Line 186: As the authors describe earlier, image processing is recommended for Wild-ID, but it is not required. This should be clearer in the table.

We have changed it, thanks. Table 1:

 Image processing required? Available for which species?

Amphibian & Reptile Wildbook (ARW) No Yellow-bellied toad (Bombina variegata), European fire salamander (Salamandra salamandra), Near-eastern fire salamander (S. infraimmaculata)

AmphIdent Yes Great crested newt (Triturus cristatus), European fire-bellied toad (B. bombina), the Yellow-bellied toad, European fire salamander, the Marbled salamander (Ambystoma opacum)

I3S pattern+ Yes Every species with individual pattern

ManderMatcher Yes Fire salamander species

Wild-ID Recommended Every species with individual pattern

Line 231: Recognition Rate should also be defined clearly here. The results are hard to interpret otherwise.

We added a definition, thanks. L 213 “We defined the Recognition Rate as the proportion of recaptures from the full dataset that was successfully identified as a match by the software.“

Line 233: Does “we used the existing FRR from Faul et al. (2022)” mean that the authors simply adopted the definitions from this paper, but recalculate these values? Or that they simply report the exact same values originally calculated in that study?

Here we report the same values from Faul et al 2022 with Wild-ID. We also used the same dataset for the further testing in the Amphibian and Reptile Wildbook. L 214 “For the larval dataset in Wild-ID, we report the existing FRR from Faul et al. (2022).”

Line 256: Should this say “FRR” instead of “FFR”? Likewise on Lines 257, 267, and several other places.

Thanks for pointing this out. That was indeed a typing error that occurred several times. We have changed it in the whole document.

Line 296: As noted earlier, I think it is critical that “Recognition Rate” (RR) is defined. I would have assumed that it means something like “the probability of identifying a true match somewhere in the list of potential matches”. However, this doesn’t appear to be possible. For example, an FRR1 of 15.12 in Wild-ID would suggest that the correct match *was* identified in the first position for ~85% of photographs. However, the RR is only 37.88. This seems to mean that my intuition is wrong about the definition of RR. I think that a better explanation of this should be a priority.

Please see comment above.

Line 337: I think the authors may consider expanding their discussion in two ways: 1) Would there be value in combining several of these methods? The authors acknowledge that manual visual inspection + software is very effective. But what about, for example, ARW + Wild-ID? Would that improve the ability to do this reliably?; and 2) This seems like a field that is likely to change quickly and dramatically with the advent of modern artificial intelligence tools. Is there reason to believe that all of these existing tools (i.e., the ones compared in this study) might be replaced in the near future? I am not asking the authors to speculate wildly, but a brief discussion of the future directions for these methods might prove useful.

We’ve added two sentences here. L 306 “Another possibility to improve reliability of the software could be to combine results from two software programmes, like Wild-ID and ARW for instance. However, different software have different requirements for the image processing and using more than one software can also be considered quite time consuming and thus not very effective.”

We’ve also added one sentence in the conclusion. We believe that the ARW will also in the future be a valuable tool as the algorithm that is used here keeps learning and getting better the more it is used. L 327 “In addition, it is likely to deliver even better results in the future as the hot spotter algorithm gets better (“learns”) with more images being matched.”

Line 374: This does not appear to satisfy the PLOS Data policy. I recommend making these underlying data available in a public repository.

Please see new data availability statement L 346 “The larval dataset is available here https://doi.org/10.6084/m9.figshare.24999353.v1, and the adult dataset is available here https://doi.org/10.6084/m9.figshare.24998690.v1.”

---

## [Editor Report · Decision Letter 1]

23 Jan 2024

Performance of different automatic photographic identification software for larvae and adults of the European fire salamander

PONE-D-23-31936R1

Dear Dr. Schulte,

We’re pleased to inform you that your manuscript has been judged scientifically suitable for publication and will be formally accepted for publication once it meets all outstanding technical requirements.

Kind regards,

Christopher Nice, Ph.D.

Academic Editor

PLOS ONE
---

## [Editor Report · Acceptance letter]

25 Mar 2024

PONE-D-23-31936R1 

PLOS ONE

Dear Dr. Schulte, 

I'm pleased to inform you that your manuscript has been deemed suitable for publication in PLOS ONE. Congratulations! Your manuscript is now being handed over to our production team.

Kind regards, 

on behalf of

Dr. Christopher Nice 

Academic Editor

PLOS ONE